# Prussian Blue Nanoparticle-Mediated Scalable Thermal Stimulation for In Vitro Neuronal Differentiation

**DOI:** 10.3390/nano12132304

**Published:** 2022-07-04

**Authors:** Stefania Blasa, Mykola Borzenkov, Valentina Pastori, Lavinia Doveri, Piersandro Pallavicini, Giuseppe Chirico, Marzia Lecchi, Maddalena Collini

**Affiliations:** 1Department of Biotechnology and Biosciences, University of Milano-Bicocca, Piazza della Scienza 2, 20126 Milan, Italy; stefania.blasa@unimib.it (S.B.); valentina.pastori@unimib.it (V.P.); 2Department of Medicine and Surgery, Nanomedicine Center, University of Milano-Bicocca, Via Alfred Nobel, 20854 Vedano al Lambro, Italy; mykola.borzenkov@unimib.it; 3Department of Chemistry, University of Pavia, Via Torquato Taramelli 12, 27100 Pavia, Italy; laviniarita.doveri01@universitadipavia.it (L.D.); piersandro.pallavicini@unipv.it (P.P.); 4Department of Physics “G. Occhialini”, University of Milano-Bicocca, Piazza della Scienza 3, 20126 Milan, Italy; giuseppe.chirico@unimib.it (G.C.); maddalena.collini@unimib.it (M.C.)

**Keywords:** nanoparticles, thermal stimulation, neuronal differentiation, neurite outgrowth, electrical activity, electrophysiology

## Abstract

Heating has recently been applied as an alternative to electrical stimulation to modulate excitability and to induce neuritogenesis and the expression of neuronal markers; however, a long-term functional differentiation has not been described so far. Here, we present the results obtained by a new approach for scalable thermal stimulation on the behavior of a model of dorsal root ganglion neurons, the F-11 cell line. Initially, we performed experiments of bulk stimulation in an incubator for different time intervals and temperatures, and significant differences in neurite elongation and in electrophysiological properties were observed in cultures exposed at 41.5 °C for 30 min. Thus, we exposed the cultures to the same temperature increase using a near-infrared laser to irradiate a disc of Prussian blue nanoparticles and poly-vinyl alcohol that we had adhered to the outer surface of the petri dish. In irradiated cells, neurites were significantly longer, and the electrophysiological properties (action potential firing frequency and spontaneous activity) were significantly increased compared to the control. These results show for the first time that a targeted thermal stimulation could induce morphological and functional neuronal differentiation and support the future application of this method as a strategy to modify neuronal behavior in vivo.

## 1. Introduction

Heating has been the subject of increasing attention in the last years for its ability to modify cell behavior and modulate the electrical activity of excitable tissues. Previous studies showed that different combinations of time and temperature, from milliseconds at high temperature to several hours/days at mild temperature, could elicit depolarizing currents or promote neuritogenesis and induce the expression of neuronal markers [1,2,3,4]. These effects are probably due to changes in cell membrane capacitance and in ion channel properties [1,5], and to the activation of differentiation pathways [3], but the underlying mechanism by which these changes occur remains so far unknown. In many studies, infrared neuronal stimulation (INS) has been applied mainly by exploiting the absorption properties of water in the infrared (IR) region [1]. Recently, the combination of near infrared (NIR) with photothermally active nanomaterials has been employed to scale the excitation wavelength to the interval 700–1000 nm, which lies in the transparent optical windows for tissues [2,6,7,8]. IR photothermally active nanomaterials are explored as mediators to convert light, as primary stimulus, to a secondary stimulus, such as heat, which can be localized to stimulate neurons [9,10,11]. These photothermal nanoparticles, alone or together with probes and conjugates, can penetrate biological tissues and can be used for hyperthermic treatments arising from either localized surface plasmon resonances or charge transfer transitions [12]. Nanoparticles are also applied in photothermal (PTT) and photobiomodulation (PBT) therapy, which have become two of the most common therapies used in the last few years [13]. However, with optothermal stimulation, internalization of nanoparticles may cause spatial distribution variability, instability, and cytotoxicity. Nanoparticles could undergo biodegradation in the cell environment and damage cell components, such as plasma membrane and intracellular organelles. They could compromise cell membrane integrity through lipid peroxidation and generate oxidative stress or inflammation, and they could have a genotoxic potential for cells. For this reason, it is important to study the cytotoxic effect of nanoparticles to choose the safest type to use for cell treatments [14].

Prussian blue nanoparticles (PBNPs) were approved in 2003 by the U.S. Food and Drug Administration (FDA) as a safe and non-toxic compound and are available commercially [15]. They consist of a coordination polymer containing Fe^3+^ hexa-coordinated by the N atoms of [Fe(CN)_6_]^2+^ and they can be soluble or insoluble, depending on the exact formulation, namely Fe_4_[Fe(CN)_6_]_3_·xH_2_O (x = 14–16, “insoluble” PBNPs) and KFe[Fe(CN)_6_]· xH_2_O (x = 1–5, “soluble” PBNPs). Soluble PBNPs show lower-dimensional crystals that reach the size typical of the mesophase, forming clear deep-blue colored colloidal solutions, whereas insoluble PBNPs form larger crystals that easily aggregate and give a precipitate. However, all these formulas correspond to the same crystal and molecular structure by X-ray diffraction [16]. PBNPs can be functionalized using several methods [17] and are used in several applications, such as antimicrobial therapy [18] or cancer treatment, often combined with a NIR laser [19]. They can convert light into heat thanks to a metal-to-metal charge transfer between Fe^2+^ and Fe^3+^ through a cyanide bridge, resulting in an intense, large absorption band with a maximum of ~700 nm; the light irradiation in the 700–750 nm region results in a thermal relaxation.

In this study, we took advantage of PBNPs, dispersed in a polymeric matrix and not in direct contact with cells, to obtain a scalable and controlled thermal stimulation to modulate neuronal behavior and properties of an in vitro model of dorsal root ganglion (DRG) neurons, the neuroblastoma F-11 cell line. The effects of this stimulation on cell morphology and electrophysiological properties enabled us to establish a reproducible protocol for optical stimulation, showing for the first time a maturation and a neuronal differentiation for a sustained period. Validation of the efficacy of this approach on more complex in vitro and in vivo models of neurodegeneration will be fundamental to translating its applicability in clinics, especially in regenerative therapies.

## 2. Materials and Methods

### 2.1. Cell Cultures

F-11 cells (mouse neuroblastoma N18TG-2 x rat DRG, ECACC Cat#08062601 RRID: CVCL_H605; [20]) were seeded at 50,000 cells/35 mm dish (Corning^®^; Sigma-Aldrich, St. Louis, MO, USA) and maintained in a serum-deprived medium to reduce proliferation. The complete composition of the medium was as previously described [21]: Dulbecco’s modified Eagle’s medium (Cat#D6546; Sigma-Aldrich, St. Louis, MO, USA), 2 mM glutamine (Sigma-Aldrich, St. Louis, MO, USA), 1% fetal bovine serum (FBS, Cat# F2442; Sigma-Aldrich, St. Louis, MO, USA), and penicillin/streptomycin (10,000 U/mL, lot#753901 ref#15140; Gibco™, Waltham, MA, USA). Twenty-four hours after seeding, the cells were thermally stimulated. The stimulation was repeated the following day, with the same parameters of duration and temperature. After each stimulation the cells were replaced in the incubator at 37 °C, in a humidified atmosphere with 5% CO_2_. The medium was replaced with fresh medium twice per week to prevent cell starvation. Cells maintained at 37 °C in the incubator were used as control. Morphological and functional analyses were performed for 8 days after seeding for both stimulated and control samples.

### 2.2. Bulk Heating Protocol

To verify whether heating could effectively induce cell differentiation, experiments were performed in bulk configuration to find out the best protocol to be applied in irradiation experiments. In particular, the temperature of the incubator (Jouan IGO150 CELLife CO_2_ Incubator, Thermo Fisher Scientific, Rodano, Italy) was increased from the standard 37 °C to higher temperatures for 10 min/day for two consecutive days. The temperatures chosen were 39 °C, 41.5 °C, or 43 °C. Another series of experiments consisted in heating the cells for 10, 20, 30, 45, or 60 min/day for two consecutive days in order to find the best time duration of the heating protocol. Each experiment was performed on two or three independent cultures. After heating, cells were maintained in an incubator at 37 °C.

### 2.3. Prussian Blue Nanoparticle Preparation

PBNPs were prepared according to the literature [16], but increasing the concentration of the reagents from 1 mM to 10 mM. Specifically, 100 mL of a 10 mM FeCl_3_ solution was mixed with 10 mM K_4_[Fe(CN)_6_] in 0.025 M citric acid and heated at 60 °C under stirring. After 1 min stirring at 60 °C, the solution was cooled at room temperature. The solution was centrifuged for 25 min at 13,000 rpm in 10 mL test tubes for purification. The centrifuged PBNP pellet was resuspended in half the original volume. The absorbance peak of the PBNP aqueous solution was evaluated with Jasco, V-570 spectrophotometer.

### 2.4. PBNP-PVA Layer Preparation

A solution was made containing 7% (vol%) poly vinyl-alcohol (PVA, average molecular weight 72,000 g mol^−1^, degree of hydrolysis 98%, Sigma-Aldrich, St. Louis, MO, USA) and 27–30% of 10 mM PBNPs. The PVA powder was dissolved in water and was maintained in an oven at 70 °C for at least one hour. Then the PBNP solution was added to the melted polymer under continuous stirring for 1 h. An amount of 70 µL of the final solution was dropped on the outer surface of the petri dishes and dried in an oven at 70 °C for at least 1 h. The PBNP–PVA patch (blue colored) covered a circular area of approximately 1 cm of diameter. A circle of the same area was drawn on the control petri dishes to compare similar size regions.

### 2.5. Irradiation Protocol by Heating Nanoparticle Layers

A Ti:Sa laser (Mai-Tai DeepSee Ti:Sapphire^®^, Spectra Physics^®^, Santa Clara, CA, USA), tunable between 690 nm and 1100 nm, was used to increase the temperature of the medium into the petri dish in correspondence of the PVA–PBNP layer. By exploiting the 720 nm wavelength, the power was chosen to reach the desired temperatures, which were defined according to the results obtained by bulk heating experiments. Temperature calibrations were made either on dry petri dishes to check the reproducibility of the layers, and on petri dishes with 2 mL of culture medium in order to determine the power needed for temperature increase. Accurate temperature measurements were performed by a thermocamera (FLIR E40, FLIR Systems Inc., Wilsonville, OR, USA) and by a needle thermocouple (Omega Engineering Ltd., Stamford, CT, USA). The laser spot size was accurately calibrated in order to match the beam size with the photothermally active area (~1 cm of diameter). To this end, a beam expander was placed on the beam path and the spot size was measured by recording the power after passing through a variable diameter iris. The final spot size was obtained by fitting the curve of power versus the iris diameter. The petri dish was placed on the sample holder and the laser beam irradiated it from below. During the irradiation, the petri dishes were maintained at 37 °C in a home-made chamber, of which the temperature was controlled by The Cube (Life Imaging Services, Basel, Switzerland).

### 2.6. Morphological Analysis

Morphology was determined by imaging the cells for 8 days from seeding. Cells were seeded on day 0, and images were taken in transmitted light mode on day 1; after a recovery of at least 2 h, they were heated for the first time. The same procedure was repeated on day 2. Electrophysiological recordings were performed on days 7 and 8. Transmitted images were acquired on a Leica SP5 microscope (Leica Microsystems, Wetzlar, Germany) with an air objective (20**×** HCX PL Fluoter, Leica Microsystems, Wetzlar, Germany). Six tiles-mode images of 775 µm × 775 µm were acquired to cover the irradiated area where the PBNP-PVA disc was present, or an equivalent area for the control and bulk heating samples, in order to acquire images of the same region for the 8 days of the experiment, and a comparative statistical analysis was achieved among the different experiments. The images were processed by FIJI ImageJ, version 2.0.0, Opensource code [22]. Neurites were manually traced, then a homemade macro, which subtracts each traced image from the raw images, was run. The new image, on which only the traced neurites were visible, was binary converted and skeletonized. The characteristics of the traced neurites were eventually extracted in a .txt file and statistical analyses were performed.

### 2.7. Electrophysiological Analysis

The functional characterization of the electrophysiological properties of F-11 cells was performed with the patch-clamp technique in the whole-cell configuration at room temperature. Before recording, culture medium was replaced by a standard extracellular solution, which contained (mM): NaCl 135, KCl 2, CaCl_2_ 2, MgCl_2_ 2, hepes 10, and glucose 5, with a pH of 7.4. The standard pipette solution contained (mM): potassium aspartate 130, NaCl 10, MgCl_2_ 2, CaCl_2_ 1.3, EGTA 10, and hepes 10, with a pH of 7.3. Recordings were acquired by the pClamp8.2 software (pClamp, RRID:SCR_011323) and the MultiClamp 700A amplifier (Axon Instruments; Molecular Devices, LLC, San Jose, CA, USA). Resting membrane potential and action potentials were monitored in the current-clamp mode. In the voltage-clamp mode, the resistance error was compensated up to 50–70%. Sodium (I**_Na_**) and potassium (I_K_) currents were recorded by applying a standard voltage protocol [21], which started from a holding potential of −60 mV, conditioned cells at −90 mV for 500 ms, and successively clamped the membrane at depolarizing test potentials in 10 mV-increments, from −80 to +40 mV. Cell conditioning at −90 mV was necessary to remove the inactivation of sodium channels; then the depolarization to test potentials increased their activation and enabled measurement of the sodium inward currents. Potassium currents were evident as outward signals that were evoked by the depolarizing test potentials. Both round-shaped cells and cells with neurite-like processes were tested for this characterization.

### 2.8. Lactate-Dehydrogenase (LDH) Assay

In order to verify whether heating induced cell stress, we tested cell viability by measuring the lactate-dehydrogenase (LDH) activity on both stimulated and control samples. LDH is a ubiquitous enzyme usually localized in the cytosol and is released into the medium by damaged cells. The samples used for the assay were maintained at −20 °C. According to the protocol [23,24], they were defrosted in ice and centrifuged at 1000 rpm for 4 min, and the supernatant was collected for the assay. The total solution volume of 1 mL was made, which contained (μL): K-phosphate buffer 850, NADH 20, and a stimulated or control sample, 70. The reaction started by adding 60 μL of pyruvate. The rate of the absorbance decrease over time was measured and the ratio of LDH activity (U/mL) in the cell culture medium was calculated using the standard formula.

### 2.9. Statistical Analysis

For the data analysis, Origin 9 (OriginPro, Version 2019, OriginLab corporation, Northampton, MA, USA) and Excel (Microsoft, Redmond, WA, USA) were used. Data are presented as mean ± standard error (S.E.). Mean comparisons were obtained using the parametric one-way ANOVA test or the non-parametric Mann–Whitney test. Percentages of cells with spontaneous electrical activity were compared using the χ^2^ test. The significance level was set for *p* ≤ 0.05.

## 3. Results

### 3.1. Effects of Bulk Heating: Morphological and Functional Characterization

Experiments in bulk heating were performed to evaluate the eventual effects on the morphology and the electrical activity of F-11 cells. This is a neuroblastoma cell line, which expresses functional properties of mature sensory neurons under appropriate culture conditions [21] or when seeded on matrices mimicking the extracellular environment [25]. Temperatures ranging from 39 °C to 43 °C and exposure timings ranging from 10 to 60 min were tested. Concerning the morphological properties, the most efficient time/temperature combination was 30 min at 41.5 °C (Figure 1 and Appendix A), which induced significant elongation in neurites (about 27% of increase on day 8). By contrast, the lowest temperature, 39 °C, induced no effect on cell morphology, whereas the highest temperature, 43 °C, seemed to determine cell stress, which became particularly evident after the fourth day, when the mean number of cells with neurites decreased at 5% of cultures maintained at 37 °C. Since a visible decrease in neurite length and an increase in cell stress/mortality was seen in samples maintained at ≥43 °C, chosen temperatures did not exceed 42 °C.

Moreover, cells at 41.5 °C showed the tendency to sprout a higher number of processes than cells under control conditions (Figure 2A). According to the results obtained by morphological analysis, an electrophysiological investigation was performed by the patch-clamp technique, both on cells with neuronal morphology and on cells with round shape, to verify whether the established protocol could also induce functional differentiation. The electrophysiological parameters investigated were the resting membrane potential, the electrical activity, and the sodium and potassium current densities. The resting membrane potential was more hyperpolarized in cells exposed at 41.5 °C compared to control cells (−42 mV ± 1 mV versus −32 mV ± 2 mV, *p* = 0.001, Mann–Whitney test). Moreover, in heated cultures, a significantly higher percentage of cells with spontaneous activity was present (67% versus 25%, *p* = 0.009, chi-square test) and a higher action potential firing frequency was measured compared to control (4.6 ± 0.7 Hz versus 2.6 ± 0.9 Hz, *p* = 0.03, Mann–Whitney test) (Figure 2B,C).

With the electrical activity, heated cells showed a consistent trend of expressing higher sodium and potassium current densities compared to cells maintained at 37 °C (Figure 3). Since these parameters are the hallmarks of neuronal maturation, these results indicate that bulk heating could also induce functional differentiation in the F-11 cell line.

To exclude the fact that this approach could induce cell stress, a lactate-dehydrogenase (LDH) assay was performed on days 7 and 8. The results, shown in Table 1, suggested that heated cells released in the medium levels of LDH equivalent to control cells (*p* = 0.94, one-way ANOVA test, *n* = 8 samples for each condition), indicating that the treatment was not detrimental to F-11 cell survival.

### 3.2. Smart Petri Dish Characterization

Considering the efficacy of bulk heating on cell differentiation, the combination of 41.5 °C for 30 min of exposure was also chosen to induce a localized and selective stimulation by means of NIR laser irradiation of PBNP-based polymer layers applied on the bottom of petri dishes (smart petri dishes). PBNPs were evenly dispersed in the hosting polymer matrix; evenness was verified by imaging the smart layer on the petri dish by means of an optical confocal microscope in reflection mode (Figure 4), from which the back-scattering of the PBNP was collected upon excitation at 633 nm. A z-scan performed on the overall extension of the layer revealed a thickness of 90 ± 10 µm in agreement with the estimate obtained with a gauge.

In order to accurately measure the temperature reached within the medium during irradiation, the exact conditions used in the NIR laser irradiation experiments were reproduced, and a needle thermocouple was mounted and fixed inside the chamber and fed into the 2 mL solution in the petri dish through a custom-drilled lid. The temperature increase was recorded versus time at different laser power levels in order to select the proper value for obtaining the requested temperature increase. Two examples of the curves obtained are shown in Figure 5 for I = 0.15 W/cm^2^ and I = 0.07 W/cm^2^. As can be inferred from the figure, after 2 min an equilibrium temperature was reached, and this was maintained throughout the irradiation time. The plateau value depended on the irradiation intensity and the nanoparticle concentration in the PBNP-based polymer layer.

### 3.3. Effects of Thermal Increase by PBNP Irradiation 

During the irradiation procedure, petri dishes were maintained in a box at 37 °C. The temperature of 41.5 °C was reached only on the PBNP-PVA disc by use of the laser beam. Cultures maintained at 37 °C in an incubator were used as control. The morphological characterization showed that irradiated cells had longer neurites compared to the control, especially on days 7 and 8 (Figure 6A,B) as previously reported in bulk heating experiments. Moreover, thermally stimulated cells showed an increase in mean neurite number starting from day 2, and especially on days 2 and 3, compared to the control (Figure 6C), suggesting that this method of thermal stimulation could induce neuronal differentiation.

A control experiment in which cells were irradiated without the support of PBNPs was performed to verify whether the use of NIR laser could induce differentiation by itself. Cells irradiated without PBNPs had a trend comparable to the control on all days of the experiment; moreover, they had shorter neurites compared to cultures irradiated with PBNPs, suggesting that the single NIR laser did not induce differentiation of F-11 cells (Appendix A).

A temperature increase up to 43 °C was also induced by laser irradiation. After being irradiated at this temperature, cells showed shorter neurites compared to the control (Appendix A) for all the examined time points. Interestingly, although the number of cells seeded on day 0 was the same in each petri dish (5 × 10^4^ cells), in 43 °C-treated samples, starting from day 4, cells attached in the middle of the irradiated PBNP-PVA disc were fewer, suggesting that this temperature caused excessive cellular stress, as previously shown in bulk heating experiments.

Irradiated cells at 41.5 °C had a typical neuronal morphology and were able to form several small neuronal networks. In order to verify the eventual development of the typical properties of electrically mature neurons, we performed an electrophysiological investigation on day 7. Irradiated cells had a resting membrane potential more hyperpolarized compared to the control (−39 ± 1 mV versus −30 ± 2 mV, *p* = 1.94 × 10^−5^, Mann–Whitney test) and a higher action potential firing frequency (5.9 ± 0.5 Hz versus 2.5 ± 0.5 Hz, *p* = 1.12 × 10^−5^, Mann–Whitney test, Figure 7A) compared to control cells. Moreover, in irradiated cultures, a significantly higher percentage of cells with spontaneous activity was found (68% versus 25%, *p* = 3 × 10^−5^, chi-square test, Figure 7A). Cells exposed to the thermal increase showed a higher sodium current density (110 ± 11 pA/pF versus 80 ± 11 pA/pF, *p* = 0.05, one-way ANOVA test) and a trend to a higher potassium current density compared to control cells (Figure 7B). These results confirm that thermal increase by irradiated PBNPs could induce a functional differentiation in F-11 cell line.

As for the cultures exposed to bulk heating, the LDH assay was performed on both irradiated and control samples on days 7 and 8. Data in Table 2 show that LDH levels in the medium of thermally stimulated cultures were not significantly different from the control, indicating that this treatment did not impair cell viability (*p* = 0.10, Mann–Whitney test, *n* = 6 samples for each condition).

## 4. Discussion

In this paper, we show the effect of a new approach for in vitro neuronal scalable thermal stimulation, constituted by irradiating PBNPs by a NIR laser. PBNPs were embedded in a PVA disc adhered to the outer surface of the petri dish in which the cells to be stimulated were maintained in culture. By this approach, we demonstrated that a temperature increase from 37 °C to 41.5 °C for 30 min, repeated for two days, was sufficient to induce neuronal differentiation in F-11 cells, an in vitro model of DRG neurons that has been used in literature for its properties consistent with those of DRG and primary neurons, and which has also been employed for tissue scaffold development for neuronal regeneration [26,27]. Neuronal differentiation was investigated and demonstrated by both morphological and functional analysis.

Cell imaging for 8 days after seeding showed that neurites tended to increase in number and were significantly longer in thermally stimulated cultures compared to the control. Moreover, electrical properties (resting membrane potential, Na^+^ and K^+^ current densities, action potential firing frequency, and spontaneous activity), recorded on days 7 and 8, also reached values characteristic of mature neurons, confirming that thermally stimulated cultures acquired a significant functional differentiation compared to control cultures. These results indicate that the new approach of thermal stimulation we propose could induce long-term modifications (maintained for at least 8 days) of neuronal properties without the support of genetics or chemical compounds.

Since the brain is one of the most temperature-sensitive organs, infrared laser has already been used to stimulate neurons, alone or in combination with nanoparticles [10,28]. However, in several previous studies, the effects obtained on neurons consisted in transient variations in cell properties (membrane depolarization, action potential firing modulation). Some articles showed significant changes in morphology [1,2,3,4] that were maintained for a few days, but no electrophysiological analysis has been performed to demonstrate functional modifications. Moreover, in articles showing the capability of thermal stimulation to induce neuronal differentiation, cell culture media were enriched with differentiating components or factors, preventing the isolation of the real efficacy of temperature increase [29].

PBNPs are versatile tools endowed with photothermal effects, and are excellent candidates for in vivo treatments due to their biocompatibility and biodegradability. PBNPs have been explored so far for imaging and chemotherapy, especially for their cancer-cell-killing ability [30], and their biostability and ability to convert light into heat facilitate their employment in the field of the regenerative medicine.

In this work, our decision to develop a polymer matrix embedded with photothermal nanoparticles, thereby obtaining a “smart” petri dish supporting cell culturing, yielded the significant advantage of preventing any contact with the cells, and allowed the design of a precise and necessary geometry for the active heating area, thereby inducing differentiation on a selected region of the cell culture. Moreover, with this approach, the irradiation temperature was able to be finely tuned and selected by a calibration of the laser spot size and power on one hand, and by the photothermal particle concentration in the smart region on the other.

The ability to induce neuronal differentiation by means of scalable heating opens new possibilities for treating peripheral nerve injuries and/or neurodegeneration. Among the techniques available, surgery, cell-based therapy, and optogenetics are the most frequently used so far. Surgery is the most common therapy, but in general, patients experiencing functional recovery after surgery do not exceed 50%, and the intervention can lead to neuronal atrophy [31]. Cell-based therapy is a promising approach, but is an invasive process, and safety cell preparations are time-consuming and expensive [31]. Optogenetics, in which neurons are genetically modified with light-sensitive ion channels, has gained great interest in the last years, but it requires gene transfection into neuronal cells, which has several limits: the expression efficiency is spatially heterogeneous, the high expression rate could lead to toxic accumulation of protein within the tissue, and the light could be refracted or absorbed by the multiple tissue layers [32]. The approach described in this paper has several advantages: it has no direct contact with cells, the preparation is fast and economic, the ability of the nanoparticles to convert the light into heat allows the use of the NIR laser at a low potency, and the stimulation parameters (laser power) and PBNP concentration can be easily modified. Moreover, the change in shape of the active region, for example, to elongated patterns, suggests that this approach could be a promising tool in the precision medicine field. From the perspective of the clinical translatability of this approach and its potential application in the biomedical field, the molecular and physiological mechanisms underlying the obtained results should be assessed, and other cellular models should be considered to verify the reproducibility of the effects described in this paper. In fact, cell lines generally reproduce the principal properties of primary cells, but they do not express all the receptors and ion channels that sustain their complex physiology [21]. In vitro models of nerve injuries already available for studying peripheral nerve regeneration, such as DRG/Schwann co-cultures, embryonic spinal cord motor neurons, pluripotent stem cells, or organotypic models [33], would provide a reliable confirmation of the efficacy of the approach and would permit the investigation of the molecular mechanisms underlying the neuronal differentiation induced by this scalable thermal stimulation technique.

## 5. Conclusions

In this paper, we show for the first time a novel method to induce neuronal differentiation by using a combination of light sensitive nanoparticles and NIR laser, which we applied with success on a neuroblastoma cell line maintained in culture without chemical differentiating agents or genetic techniques, and without any contamination by the photothermal material. The smart matrix delivering heat can be tuned to the temperature/shape needed for the region to be stimulated. These results show that targeted heating could be a promising approach for in vivo therapy to induce neurite outgrowth and neuronal behavior modifications.

## 6. Patents

Stefania Blasa, Mykola Borzenkov, Piersandro Pallavicini, Maddalena Collini, Giuseppe Chirico, and Marzia Lecchi are the inventors of the nanoparticle-near-infrared laser technology that induces cell differentiation mentioned in this publication. The patent application was filed by Botti & Ferrari S.p.A.

## Figures and Tables

**Figure 1 nanomaterials-12-02304-f001:**
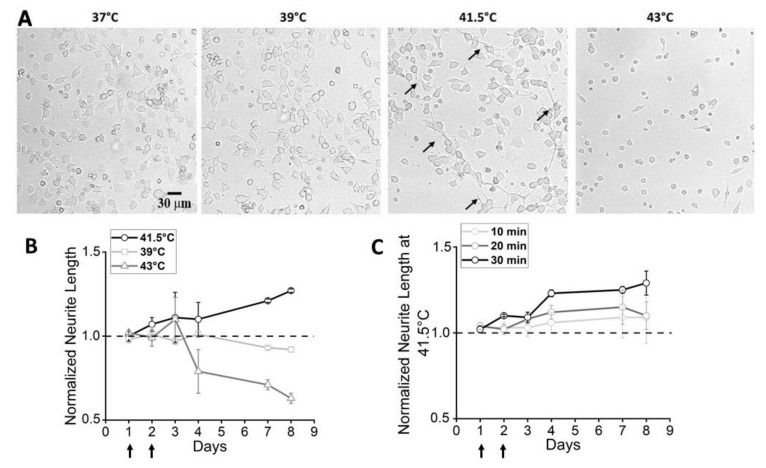
Morphological properties of F-11 cells exposed to bulk heating. (**A**) Representative images of cells at day 8, after exposure to bulk heating for 30 min at different temperatures. At 39 °C, cells did not show differences with control cells (37 °C), whereas, at 41.5 °C, an increase in neurite length (black arrows) was evident. Temperatures higher than 41.5 °C (43 °C) had detrimental effects on cell viability. (**B**) Neurite length normalized to the value of the control group one day after seeding. The different symbols refer to the different temperatures applied for 30 min on day 1 and day 2 after seeding (indicated by black arrows). The temperature of 39 °C (squares) did not induce any effect on cell morphology compared to the control. In the 41.5 °C samples (circles) neurites were longer versus the control, starting from day 3. The temperature of 43 °C (triangles) seemed to induce cell stress, particularly from day 4. (**C**) Normalized neurite length of heated (41.5 °C) F-11 cells at different times of exposure. Neurite of cells maintained at 41.5 °C increased in length with longer exposure times.

**Figure 2 nanomaterials-12-02304-f002:**
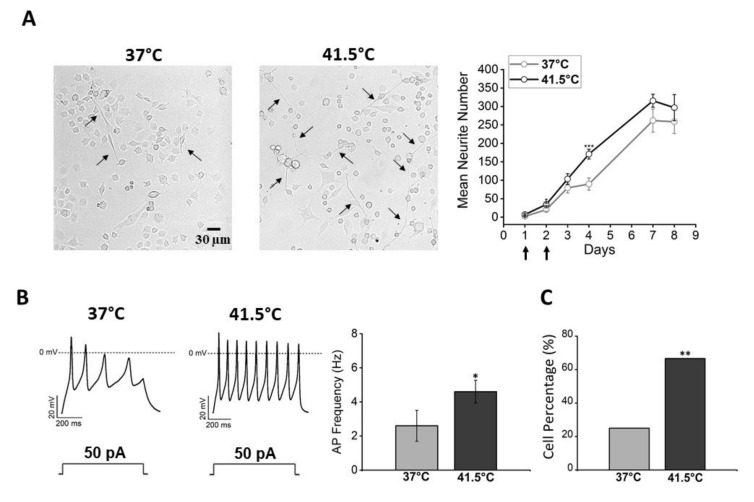
Morphological and electrophysiological properties of F-11 cells thermally stimulated at 41.5 °C for 30 min. (**A**) Cells heated at 41.5 °C on day 1 and day 2 after seeding showed more numerous (mean neurite number/imaged area) and longer neurites (black arrows in the picture) compared to control cells maintained at 37 °C, and (**B**) showed the ability to discharge spontaneous or induced action potentials at a higher frequency compared to cells maintained at 37 °C (for 41.5 °C samples: 4.6 ± 0.7 Hz, *n* = 22; for 37 °C samples: 2.6 ± 0.9 Hz, *n* = 15; *p* = 0.03, Mann–Whitney test). (**C**) Moreover, 41.5 °C samples had a higher percentage of cells with spontaneous activity compared to the control, indicating that thermally stimulated cultures had a higher probability of developing small neuronal networks (for 41.5 °C samples: 67%, *n* = 16/24; for 37 °C samples: 25%, *n* = 4/16, *p* = 0.009, chi-square Test). * *p* ≤ 0.1, ** *p* < 0.01 and *** *p* < 0.001.

**Figure 3 nanomaterials-12-02304-f003:**
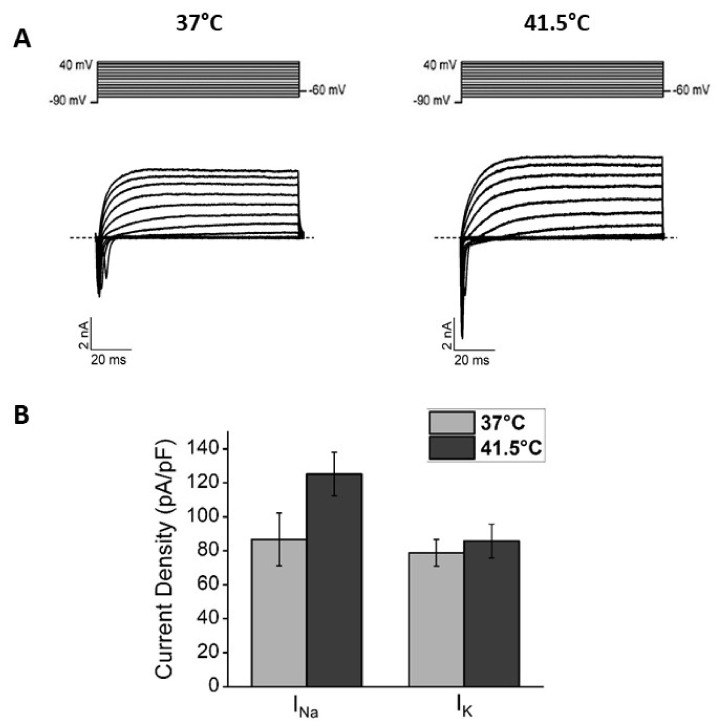
Functional analysis of F-11 cell electrophysiological properties by the patch-clamp technique on days 7 and 8 showed a more differentiated profile for heated cells versus 37 °C cells. (**A**) Representative sodium and potassium current traces obtained by the protocol indicated above and described in the Materials and Methods section. (**B**) Current density bar graphs; heated cultures showed a trend to express higher sodium and potassium current densities compared to the control (for 41.5 °C samples: I_Na_, 125 ± 13 pA/pF and I_K_, 86 ± 10 pA/pF, *n* = 23; for 37 °C samples: I_Na_, 87 ± 16 pA/pF and I_K_, 79 ± 8 pA/pF, *n* = 15; for I_Na_ *p* = 0.11, Mann–Whitney test, for I_K_ *p* = 0.9, Mann–Whitney test).

**Figure 4 nanomaterials-12-02304-f004:**
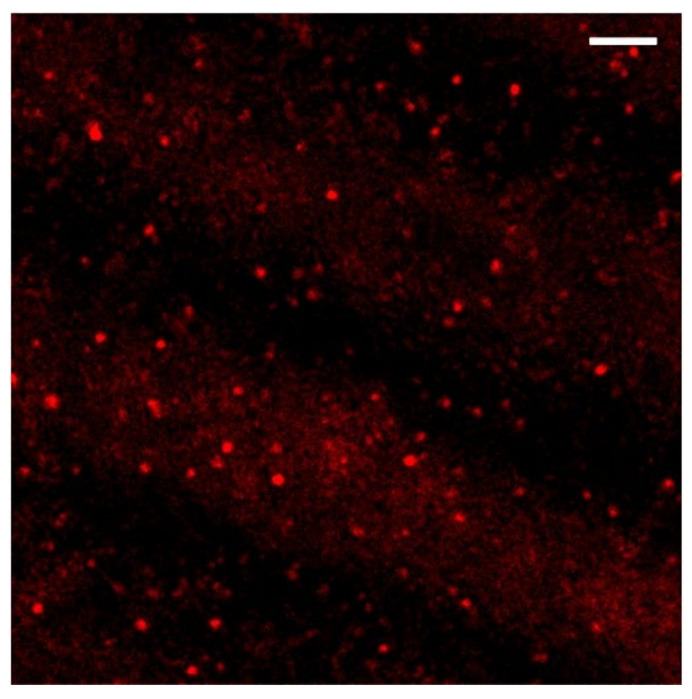
A 30 µm × 30 µm field of view of a selected plane in a z-scan acquisition of the PBNP-PVA smart layer (the bar corresponds to 3 µm). The red spots represent the reflection signal of the PBNPs upon 633 nm excitation.

**Figure 5 nanomaterials-12-02304-f005:**
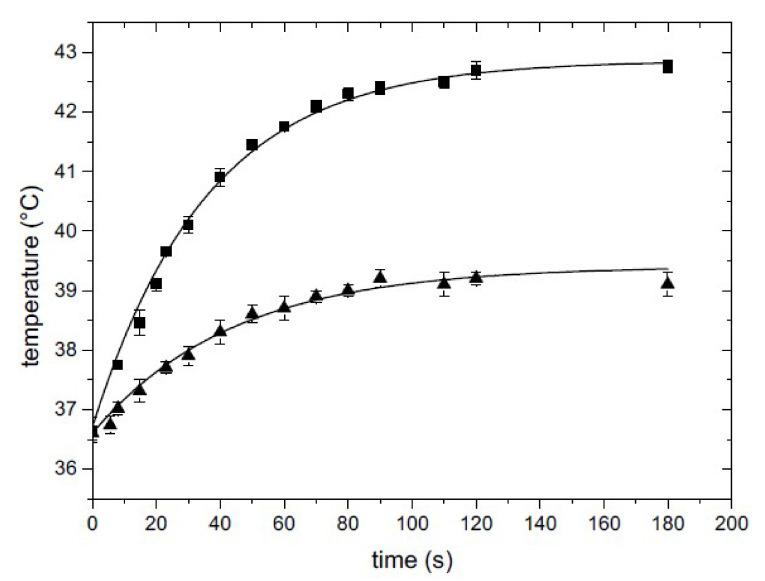
Temperature increases versus irradiation time measured by a needle thermocouple directly inserted in the Petri dish with 2 mL of solution. Squares I = 0.15 W/cm^2^, triangles I = 0.07 W/cm^2^.

**Figure 6 nanomaterials-12-02304-f006:**
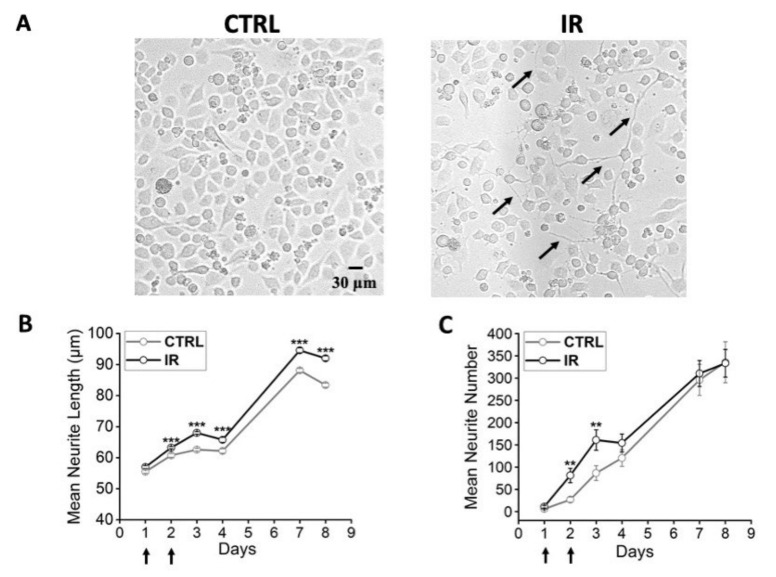
Morphological properties of irradiated F-11 cells. (**A**) Representative images show longer and more numerous neurites in cultures irradiated at 41.5 °C (IR) than in cultures maintained at 37 °C (CTRL). (**B**) Neurite length of cells irradiated at 41.5 °C on day 1 and day 2 after seeding (indicated by black arrows), analyzed from day 1 to day 8, was significantly longer than the control starting from day 2 (*p* < 0.001, Mann–Whitney test). (**C**) In the irradiated cultures, the number of neurites in the imaged area increased starting from day 2, with a significant difference on day 2 and 3 (*p* = 0.002 and *p* = 0.01, respectively, Mann–Whitney test). Data were collected from 6 experiments for each condition, and the number of samples used in each experiment was 4. ** *p* ≤ 0.01 and *** *p* < 0.001.

**Figure 7 nanomaterials-12-02304-f007:**
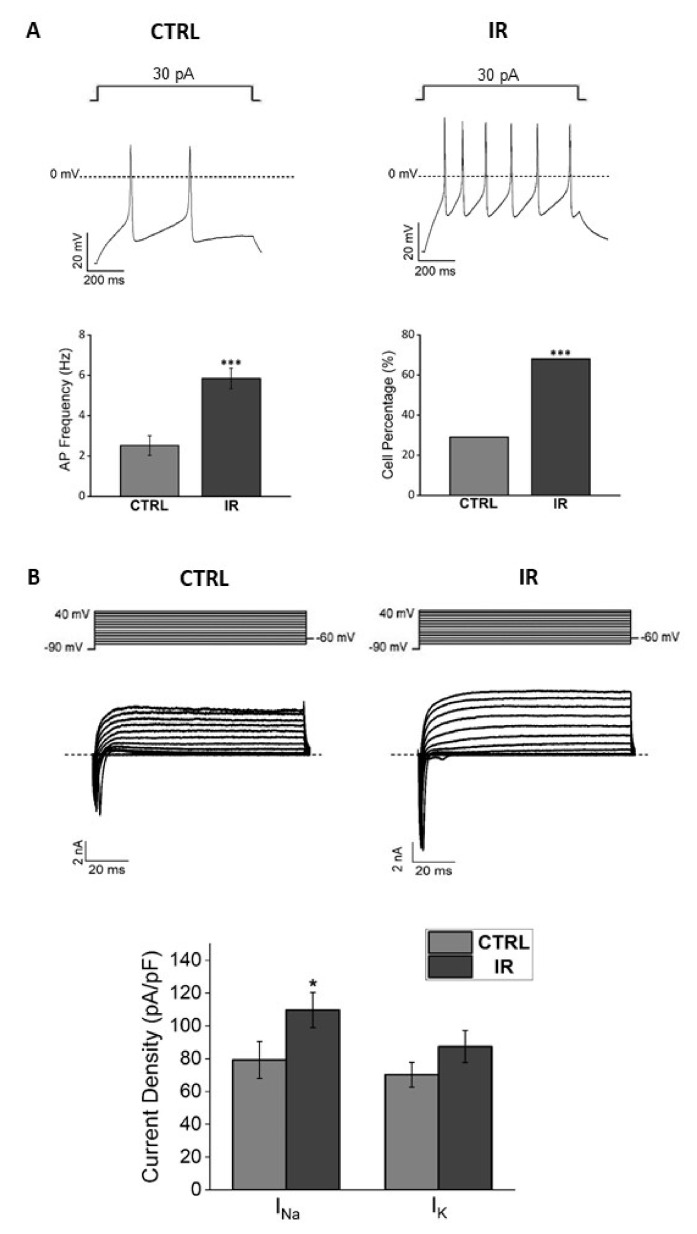
Functional analysis of F-11 cell electrophysiological properties by the patch-clamp technique on days 7 and 8 confirmed a more differentiated profile for stimulated cells (IR) versus 37 °C (CTRL). (**A**) Irradiated cultures had higher action potential firing frequency (for irradiated samples: 5.9 ± 0.5 Hz, *n* = 36; for control samples: 2.5 ± 0.5 Hz, *n* = 36; *p* = 1.12 × 10^−5^, Mann–Whitney test) and a higher percentage of cells with spontaneous activity compared to the control (for irradiated samples: 68%, *n* = 25/37; for control samples: 25%, *n* = 7/36; *p* = 3.42 × 10^−5^, chi-square test). (**B**) Stimulated cultures showed a higher sodium current density and a trend to had higher potassium current density compared to the control (for irradiated samples: I_Na_, 110 ± 11 pA/pF, and I_K_, 87 ± 10 pA/pF, *n* = 32; for control samples: I_Na_, 80 ± 11 pA/pF and I_K_, 70 ± 7 pA/pF, *n* = 31; for I_Na_ *p* = 0.05, one-way ANOVA test, for I_K_ *p* = 0.3, Mann–Whitney test). In the bar graphs: * *p* ≤ 0.05 and *** *p* < 0.001.

**Table 1 nanomaterials-12-02304-t001:** Lactate-dehydrogenase activity measured on bulk heated cells versus control samples. Results showed that this treatment had no detrimental effects on heated samples (*p* = 0.94, one-way ANOVA test, *n* = 8 for each condition).

Bulk Heating (Days 7–8)			
**37 °C**	**S.E.**	**41.5 °C**	**S.E.**
4.98 × 10^−8^	1.28 × 10^−8^	4.86 × 10^−8^	1.16 × 10^−8^

**Table 2 nanomaterials-12-02304-t002:** Lactate-dehydrogenase activity measured on thermally stimulated cells versus control samples. Results showed that thermal stimulation did not impair cell viability in 41.5 °C cells (*p* = 0.10, Mann–Whitney test, *n* = 6 for each condition).

Thermal Stimulation (Days 7–8)			
**CTRL**	**S.E.**	**IR**	**S.E.**
9 × 10^−8^	1.2 × 10^−8^	15 × 10^−8^	2.3 × 10^−8^

## Data Availability

Not applicable.

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
