# Peer review of "Prussian Blue Nanoparticle-Mediated Scalable Thermal Stimulation for In Vitro Neuronal Differentiation"

_nanomaterials, 2022, doi:10.3390/nano12132304_

Round 1

Reviewer 1 Report

This manuscript describes a new approach for scalable thermal stimulation to modulate neuronal structure and function of an in vitro model of dorsal root ganglions (F-11 cell line). The authors provide interesting results, which demonstrate the effectiveness of the method. The topic is interesting in itself, and the article seems to have some potential to be attractive to readers interested in biological applications of nanomaterials. It is suitable for publication in Nanomaterials with some minor amendments.

General comments:

In the Introduction (and paragraph 3 of Discussion), it appears that none of the previous work has demonstrated both differentiation (structure) AND electrical activity (function) in response to photothermal stimulation in the same tissue culture model. If this is the first such report, it may be worth flagging this achievement more positively e.g. in Abstract.

A significant element of this manuscript is that Prussian Blue nanoparticles are considered safe and non-toxic for use on humans. What are the disadvantages of these materials? The limited absorption range seems to be one disadvantage, compared to say gold nanoparticles which can be tuned to cover a much wider range of absorption wavelength. Chemical functionalisation is presumably also more challenging compared to the well-established surface chemistries that can be applied to some other nanoparticles.

I note that the water soluble PBNP-PVA patch is placed on the outside of the petri dishes. At a fundamental level, this raises the question whether it would be simpler to use a conventional heating mat or equivalent to control the culture temperature. This issue is addressed to some extent in Paragraph 5 of the Discussion, but in reality I think that the real benefit of the PBNP approach may be for regenerative therapies, as discussed in Par. 6 (and the Abstract). In particular, the current in vitro study provides some proof-of-concept before approaching more complex in vitro and in vivo neurodegenerative models. I suggest to include a few sentences to this effect in the Introduction as well, to ensure that the underlying motivation is clear from the start.

The Graphical Abstract is quite “busy”. Will the detail be visible at the published scale?

Some specific comments: 

Section 2.4: percentage composition, is this based on mass or volume?

Laser heating protocol: it is known that the high intensity, ultrafast pulses from Ti:Sa lasers can lead to relatively large temperature spikes close to the absorbing nanoparticle. It is unlikely that the monitoring techniques employed here would be able to detect these transients. While these transients don’t appear to have caused any particular issues in the reported work, please comment on the choice of this laser as opposed to more widely accessible laser diodes, for example.

Section 2.7: please provide a reference for the standard voltage protocol for measuring sodium and potassium currents. Can the underlying principle be described briefly in the text?

Section 2.8: please provide more details/references or the protocol, supplier and standard formula.

Figures 2 and 3: is spontaneous neuronal activity reliably associated with network activity? Please provide a reference for this statement. Similarly, please provide a reference for the statement that higher sodium and potassium current densities are hallmarks of neuronal maturation. Please plot the sodium and potassium current traces in Figure 3A on the same scale, or justify the use of different scales.

Figure 4: I worked out that Prussian Blue is not fluorescent, but it may be worth mentioning that the confocal imaging used backscattered light at 633 nm. Is it possible to increase the resolution of this image?

Figure 5: what laser intensity (irradiance) was used to achieve 41.5 °C under culture conditions? Please capitalise the axis titles and check for consistency in all figures.

Section 3.3: Figure 6C is mentioned in the text, but not in the figure itself. Overall the increase in mean neurite length and mean neurite number seems relatively modest compared to the images in Figure 6A. I would suggest to select images that are more indicative of the quantitative data. Given that the most significant changes seem to occur within the 24 hrs immediately after the irradiation, have you considered extending the irradiation over a longer period. Please discuss in the text.

Figure 7B: there appears to be a disagreement between the graph and the text for the percentage of cells with spontaneous activity compared to the control samples (19% vs ~28%?).

Conclusion. Paragraph 2: please provide reference(s) for values characteristic of electrical properties for mature neurons (resting membrane potential, Na+ and K+ current densities, action potential firing frequency and spontaneous activity).

Par. 3: please provide references for statement “Some articles showed significant changes in the morphology…”

Some other minor concerns: check (and use consistently) the journal conventions for decimal comma/point, as well as comma for digit grouping e.g. 13,000 rpm; 20X objective = 20×; D = Germany; S.E. = standard error; subscripts/superscripts for Chi-squared, IK and INa; remove red underlining from titles in Tables 1 and 2. 

Author Response

Please see the attachment for the responses to Reviewer 1.

Reviewer 2 Report

The manuscript propose a new method for nanoparticle-based thermal stimulation of neurons. Based on the Prussian blue nanoparticle, which is proven for its biocompatibility, the technique is applied to stimulate neuron-like F011 cell line. It is worthwhile to be published since it propose a novel method with optimized experimental conditions in terms of temperature and synthesis of the materials. Overall, the manuscript describe the topic logically and systematically. However, I’d like to ask authors to improve the manuscript regarding the following issues.

1)     In order to prove the feasibility of the proposed method, the authors used neuroblastoma F-11 cell lines. In general, the cell lines have different characteristics from the neuronal cells in the nerve system especially in terms of electrophysiological properties. I believe it is needed to explain how similar F-11 cell lines to the neurons. It would be nice to introduce previous literatures in which the same cell lines is used to mimic the primary neurons.

2)     When the PVA-PBNP layer is prepared in the petri dishes, it appears that the control condition used the petri dished without any PVA-PBNP. (In line 131, the control condition is described) In order to directly show the effect of thermal treatment with laser, the control condition should use the same preparation with the PVA-PBNP prepared samples. Please make sure about this issue.

3)     In the result, line 213-215, it states that decrease in neurite length and increase in cell mortality was seen at the temperature above 43 Celius degree. It would be nice to include the data in Figure 1.

4)     Figure 2 and 3 shows that the electrophysiological properties are changed after the temperature modulation. I’m wondering what the criteria to select the cells for patch clamp experiments were. Because the figure shows not all the cells showed the increased neurite growth, the result might be different if random cells were selected or only morphologically changed cells were selected.

5)     In discussion, authors mentioned that the applicability of the proposed method. However, it is not clear how this method can be used for clinical purposes. Please explain the details of the clinical translatability of the proposed method.

Minor issues

1)     Affiliation numbering is incorrect. Please correct.

2)     Throughout the manuscript, the decimal point is expressed with ‘,’ (comma). Is it okay with the journal publication? I know it is common in European countries but it seems strange in scientific article. Please check with the journal editor.

3)     In figure 2 (A), the y-axis of the graph is mean neurite number. I think the number is counted in the specific region of the microscope view. I suggest to express the mean neurite number per unit area.

4)     The tables in the manuscript seems quite crude. Sometimes it has the error lines, probably from the word process software. Please remake the tables.

Author Response

Please see the attachment for the responses to Reviewer 2.

Reviewer 3 Report

Nice manuscript showing PBNPs/NIR stimulation as a neuronal differentiarion inducer. Please, elaborate further the potential limitations of F11 cells.

Figure 1 data might be part of supplementary materials. Fig 1 lacks of statistical analysis as it is representative of others experiments. 

Figure 2. Please, indicate actual p values instead of <. The sample size is confusing. What the authors mean when they state n=4/16? Are these number related to patch clamp experiments?

Missed Figure 6C?

Figure 6. Sample experimental size?

Table 2. What is n=6 for the authors?

PBNPs

showing for the first time a maturation and a long-time 85 maintained neuronal differentiation.

Author Response

Please see the attachment for the responses to Reviewer 3.
